# ACTH-dependent Hypercortisolemia in a Patient with a Pituitary Microadenoma and an Atypical Carcinoid Tumour of the Thymus

**DOI:** 10.3390/medicina55120759

**Published:** 2019-11-27

**Authors:** Angelika Baranowska-Jurkun, Magdalena Szychlińska, Wojciech Matuszewski, Robert Modzelewski, Elżbieta Bandurska-Stankiewicz

**Affiliations:** Clinic of Endocrinology, Diabetology and Internal Medicine, Department of Internal Medicine, School of Medicine, Collegium Medicum, University of Warmia and Mazury, Olsztyn, Żołnierska 18, 10-561 Olsztyn, Poland; magdaaa4@o2.pl (M.S.); wmatuszewski82@wp.pl (W.M.); robur23@interia.eu (R.M.); bandurska.endo@gmail.com (E.B.-S.)

**Keywords:** ACTH-dependent hypercortisolemia, atypical thymic carcinoid, pituitary microadenoma, paraneoplastic syndrome

## Abstract

Cushing’s syndrome (CS) is a set of clinical symptoms which occur as a result of hypercortisolemia. Endogenous ACTH-dependent CS related to an ectopic ACTH-secreting tumour constitutes 12%–17% of CS cases and is one of the most common causes of paraneoplastic syndromes. This study presents a case of a 31 year-old man with diabetes, hypertension, rosacea, purple stretch marks and hypokalemia. Findings of diagnostic procedures include high concentrations of cortisol and ACTH, pituitary microadenoma and a tumour in the anterior mediastinum. Dynamic hormone tests determined the source of excess hormone secretion and ectopic ACTH-dependent CS was diagnosed. Due to increasing symptoms of superior vena cava syndrome, an emergency resection of almost the whole tumour was performed, with only a small part of the upper pole left because of the proximity of large vessels and a risk of damaging them. On the basis of histopathological tests, an atypical carcinoid tumour of the thymus was identified. Immediately after the surgical procedure, there was a significant reduction of clinical and laboratory traits of hypercortisolemia, yet, during the 46 weeks of postoperative observation, despite chemotherapy, the progression of residual masses of the tumour occurred with metastases and increased hormone indices. The presented case shows and discusses the differentiation of ACTH-dependent hypercortisolemia and its causes, difficulties in surgical therapy and chemotherapy, as well as prognosis for atypical carcinoid of the thymus, which is a rare disease.

## 1. Introduction

Cushing’s syndrome (CS) is a set of clinical symptoms which occur as a result of hypercortisolemia. An excessive concentration of cortisol in the body can have either exogenous or endogenous aetiology. In most cases, its origin is iatrogenic. The incidence of this rare disease is 0.7–2.4 cases in a million of the population in a year [1,2]. There are two forms of endogenous CS: ACTH-dependent, caused by excessive secretion of ACTH, and ACTH-independent, caused by autonomous hyperactivity of the adrenal cortex. The co-occurrence of an increased concentration of ACTH and hypercortisolemia can be caused by pituitary adenoma, ACTH- or CRH-secreting tumour [3]. Ectopic syndrome related to an ACTH-secreting tumour accounts for 12%–17% of CS cases [4] and is among the most common causes of paraneoplastic syndromes [5]. Clinical evaluation, diagnostics and treatment of patients with endogenous CS constitute a significant endocrinological problem.

## 2. Case study

A smoking 31-year-old man with 1.5-year history of poorly controlled hypertension treated with angiotensin inhibitor and calcium channel blockers and a 6-month history of diabetes with metabolic imbalance treated with intensive functional insulin therapy and with metformin was admitted to the Clinic of Endocrinology, Diabetology and Internal Medicine of the Regional Specialist Hospital in Olsztyn. The patient was admitted as he manifested clinical and laboratory features of hypercortisolemia and body weight loss of about 18 kg in 6 months, swelling of lower limbs, decreased muscular strength/power, mood changes and lower back pain.

The physical examination revealed normal body built, BMI 20 kg/m^2^, WHR 0.86, conjunctival hyperaemia, oedema, erythema and lividity of the skin on the neck and face, dilated neck veins, spread papulopustular rosacea, purple stretch marks on the hips and thighs, amyotrophy of proximal limb muscles (Figure 1). Peripheral lymph nodes accessible to palpation were not enlarged.

Laboratory tests showed lymphopenia, hypokalemia despite oral and parenteral supplementation, high concentrations of ACTH and cortisol with a rigid circadian rhythm, no suppression of cortisol secretion in a 2 mg dexamethasone overnight suppression test, a significantly increased concentration of free cortisol in 24-h urine collection and a very high concentration of chromogranin A and a lowered concentration of 5-hydroxyindoleacetic acid in 24-h urine collection (Table 1). In the CRH test, no significant increase of cortisol and ACTH concentrations occurred (Scheme 1).

A CT scan of the chest, in the anterior mediastinum from the thymus area down revealed a polycyclic smooth-contoured tumour 90 × 66 × 140 mm with centrally visible calcifications reaching the sternal notch, towards the back modelling vascular structures of the mediastinum and heart within the lateral wall of the right atrium and right ventricle. In a CT scan of the abdominal cavity and the head, no changes of metastatic character were revealed and the pancreas was normal. In an MRI scan of the pituitary gland, a hypointense oval focal lesion 5 × 3 × 3 mm in the posterior part of the anterior lobe was found (Figure 2). The ECG tests showed a slight dominance of the chambers of the right heart, without traits of overburdening and compression on the free wall of the right ventricle and right atrium, and no blockage to the inflow to the right heart, EF 67%. In a densitometric analysis of the femoral neck, the T-score for the was Neck: −3.26, and T-score Total: −2.31.

Owing to the increase of symptoms of superior vena cava syndrome, an emergency resection of almost whole tumour was performed, with only a small part of the upper pole left because of the proximity of large vessels and a risk of damaging them. On the basis of histopathological tests, atypical carcinoid tumour of the thymus was identified (Table 2). 

On the second day after the operation, the findings included a normal concentration of ACTH, normal glycemia in the 24-h profile, with no need to implement antihyperglycemic treatment, and mild hypokalemia. 

Laboratory tests six weeks after the operation included normokalemia, normal oral glucose test tolerance, and normal cortisol response to the synthetic ACTH stimulation test. In addition to the above-mentioned findings, there was hypercalcemia, hyperphosphatemia, hypoparathyroidism and vitamin D (Table 3). A scintigraphy scan of the skeleton (MDP-Tc99m) revealed increased collection of the marker in the projection of the left femoral diaphysis, which could correspond to a benign lesion, yet metastases could not be excluded (poorly defined oval shape 17 × 11 mm with increased radiolucency in the closer part of the femoral diaphysis revealed in the X-ray).

A CT scan of the chest revealed multiple very small focal lesions in the right lung, residual masses of the tumour in the anterior mediastinum at the level of the aortic arch, single lymph nodes in the mediastinum and in the right hilum, quite numerous armpit lymph nodes (Figure 3). A PET-CT scan showed features of a proliferative process with a moderately increased FDG uptake within the left supraclavicular lymph node and nodes in the mediastinum, within neck skin nodules, the chest, the lumbar region and a focus behind the sternum below the zygomatic notch of the sternum. A WB (whole body) SPECT-CT scan of somatostatin receptors centred on the lower head, neck and chest as well as the abdominal cavity and pelvis (99 mTc HYNIC TOC, 670 MBq) showed the following: in the projection of the residual focal lesion located in the anterior mediastinum, no visible significant accumulation of the radiographic marker, no other visible lesions with overexpression of SST receptors. A CT scan revealed thickened adrenal glands, especially the left one.

An ultrasound examination of the neck revealed no lesions suspected of adenoma of parathyroid glands. A fine needle biopsy of the left supraclavicular lymph node 13 × 7 mm with increased vascularity was performed, which revealed no visible fatty hilum, quite numerous calcifications up to 1 mm. The cytological material indicated a metastasis of a neuroendocrine tumour.

Considering the primary location of the tumour, its dynamics, histopathological texts, the range of operation, no expression of SST receptors and suspected metastases, following an oncologist’s consultation in patient, the treatment included chemotherapy acc. to the CAPTEN scheme (temozolomide + capecitabine) [6]. After four courses, due to the progression of the disease visible in CT scans, the current treatment was discontinued and everolimus therapy was administered [7]. Fourty-six weeks after the operation, increased concentrations of hormone markers were observed (Table 4), at the same time, an MRI scan of the pituitary gland did not show any visible progression. That is the reason why the 5-year survival rate for the 31 year-old patient was estimated to be between 56% and 77% [8], and the 10-year survival rate is as low as 9%–20% [9]. The patient has given written informed consent for publication and the use of accompanying images in this case report.

## 3. Discussion

The first descriptions of CS concerning ACTH-secreting tumours not located in the pituitary gland come from 1928 [10]. The term of ectopic ACTH secretion syndrome (EAS) was first proposed by Liddle in 1963, mainly in reference to patients with small cell lung cancer (SCLC) [11]. In the subsequent years, the aetiology of EAS underwent a considerable change, and apart from SCLC, other neoplasms whose prevalence increased decidedly were described [3]. Over 50% of CS cases of ectopic aetiology are tumours localised in the lungs: SCLC and neuroendocrine tumours of the bronchi. The ectopic ACTH secretion syndrome is much less often caused by neoplasms of the pancreas, abdomen, thymus, intestine, appendix, ovaries, thyroid gland and pheochromocytoma [12].

The patient was diagnosed with ACTH-dependent CS on the basis of hormone tests: high concentrations of ACTH and cortisol with a rigid circadian rhythm and no suppression of cortisol secretion in 2 mg dexamethasone suppression test. Imaging tests showed both a tumour of the anterior lobe of the pituitary gland as well as a massive tumour in the mediastinum. Both lesions could have led to hypercortisolemia, thus, differentiation of ectopic and pituitary overproduction of ACTH was made. Both very high concentrations of ACTH as well as no response in the CRH test unambiguously confirmed ectopic ACTH secretion. Moreover, a significantly increased concentration of chromogranin A suggested neuroendocrine nature of the tumour in the mediastinum, which was confirmed in the histopathological tests of postoperative material, in which traits of an atypical carcinoid of the thymus were found. The ACTH ectopy was ultimately confirmed by normalization of its concentration on the second day after the operation. 

A neurogenic tumour of the thymus is a rare malignant primary carcinoma of the thymus [13] which takes the form of a mass in the anterior mediastinum. The average age of patients is 42–56.6 years of age, and the tumour occurs mainly in males [14,15]. It was first described in 1972 by Rosai and Higa [16]. Four types of neuroendocrine tumours of the thymus have been distinguished: typical carcinoid of the thymus, atypical carcinoid of the thymus, neuroendocrine multicellular carcinoma, small cell carcinoma [17]. One third of all cases do not have any manifestation—lesions are revealed accidentally during routine chest X-ray examinations. Tumours can manifest symptoms caused by repositioning of or pressure exerted on mediastinal structures (pain in the chest, coughing, shallow breath, superior vena cava syndrome), in the form of distant metastases to the liver, lungs, pancreas, pleura, bones (20% of patients) or as a result of endocrinopathy (50% of the tumours are hormonally active, about 30%–40% of them are CS, 19%–25% is MEN1) [18,19,20,21].

A CT scan of the chest with contrast is a test of choice in the case of neuroendocrine tumours of the thymus. It reveals a large irregular or patchy mass in the anterior mediastinum, with a heterogenous contrast enhancement and foci of necrosis in the central section of the tumour, at times with microcalcifications. Histological differentiation of neuroendocrine tumours of the thymus includes three categories: low grade of malignancy (well differentiated G1 neoplasms), intermediate grade of malignancy (moderately differentiated G2 neoplasms), and high grade of malignancy (poorly differentiated G3 neoplasms). The majority of thymic tumours belong to the group of well differentiated neoplasms [22]. Immunohistochemical analysis plays an important role; it employs such markers as chromogranin A, NSE, CD56, synaptophysin [23]. Differential diagnosis encompasses thymoma, gangliocytoma, lymphoma, teratoma, parathyroid adenoma, parathyroid carcinoid, and medullary thyroid cancer.

Until 2016, the total number of cases of neuroendocrine tumours of the thymus was about 400 [22], while atypical carcinoid tumour is extremely rare, with an annual incidence of 0.18/1,000,000 people [24]. To date, the total number of atypical carcinoid tumours reported internationally is just over 100 [25]. These neoplasms progress aggressively, with local recurrences and metastases in 20%–30% of patients (bones, chest wall, lungs, liver, brain) [22].

The treatment of choice in cases like the one presented here is early total operative resection [15] which, in this particular case, was performed to the greatest extent possible. With the highly malignant nature of the tumour, complete surgical success can be achieved only in a limited number of cases. The long-term survival rate was 81.8% versus 9.1% without complete excision of the tumor [26]. According to other studies in patients who underwent surgical resection, the survival rate was 58% versus 26% in patient who did not accept surgery [27]. A complementary treatment consists in chemotherapy, radiotherapy, and in the case of expression of somatostatin receptors, SST receptors analogues (Y 90-DOTA-tyr3-octreotyd, 177-Lu-Dotatate) [28]. 

## 4. Conclusions

ACTH-dependent hypercortisolemia requires deep diagnostics to know the real aetiology of the disease. When diagnosing ectopic ACTH secretion syndrome, it should be remembered that its cause may be very rarely occurring atypical carcinoid tumour of the thymus—a neoplasma with extremely poor prognosis. The procedure of choice is complete surgery, which is only possible in some cases. Adjuvant treatment is chemotherapy, radiotherapy, and in the case of expression of somatostatin receptors, SST receptor analogues. The grade of histological malignancy, the size of the tumour, non-complete resection, no expression of SST receptors, distant metastases, progression of residual mass of the tumour and concentrations of particular hormones can be additional very unfavourable prognostic factors, as in the presented case.

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
