# Peer review of "ACTH-dependent Hypercortisolemia in a Patient with a Pituitary Microadenoma and an Atypical Carcinoid Tumour of the Thymus"

_medicina, 2019, doi:10.3390/medicina55120759_

Round 1

Reviewer 1 Report

Requires some English revision still:

Line 42: A smoking 31-year-old-man with 1.5-year history of badly controlled hypertension treated: (Badly)

Line 91: 6 weeks after the operation (paragraph start with Six)

Line 105: in the mediastinum and in the right hilum, quite numerous armpit lymph nodes : (axillary)

Line 182: Long-term survival rate 81.8% vs 9.1% without complete excision [22]. According to other studies: in patients who underwent surgical resection 58% vs 26% in patients who did not accept surgery [23].

Line 128: These are the reasons the 5-year survival rate for the 31-year-old patient is estimated between 56 and 77% [27], and the 10 year survival rate is as little as 9-30% [28].

Author Response

Dear Review,

Thank you very much for the review. I am sending the revised manuscript after English language corrections .Your sincerely,
Angelika Baranowska-Jurkun

Reviewer 2 Report

I believe that this work can be published in the current format.

Author Response

Dear Review,

Thank you very much for the review. I am sending the revised manuscript after English language corrections .Your sincerely,
Angelika Baranowska-Jurkun

This manuscript is a resubmission of an earlier submission. The following is a list of the peer review reports and author responses from that submission.

Round 1

Reviewer 1 Report

The case report presented by the authors is worth studying since it is rare and at se same time very interesting. It also helps to deepen the roots of the aetiology of some cases of Cushing’s syndrome (CS) caused by excessive secretion of ACTH.

The diffusion of the right diagnosis is very important but the description in this article needs some changes and clarifications.

In the first place, the form of presentation of the tables is not the most appropriate.

In table 1 numbering is improperly, the table is not arranged according to the title.

The data in the table 1refereed to free cortisol in 24-h urine collection [ug/24h] need to describe what the difference with the line follow is.

The data of cortisol and ACTH in dynamic test with CRH would be best presented as a graph.
Initial blood glucose values is assumed to be after drug treatment with
insulin and metformin (but not it said)

Also in table 1 chromogranin A data should be reviewed, it value does not agree with the text neither the table 4.

Table 3 does not have a correct description of the data according to its title.

Reviewer 2 Report

This is an very dramatic and sad case of Thymic CA, as it has such a poor prognosis: so it is important to share this case.

A very important message that has not been discussed enough in the discussion if this patient has MEN-1, and whether genetic testing has been considered. Although he has no evidence of hyperparathyroidism, this still needs to be addressed in the discussion as there may be familial implications if the index case is MEN-1 positive. 

Also i this regard to MEN-1, Thymic cancer only occurs in 2% of MEN-1 cases (so are very rare) and guidelines suggest that screening for Thymic cancer occurs every 1-2 years: although this has absolutely NO evidence base, as this is extremely rare and rapidly progressive

Line 58: specify the type of 2mg Dexamethasone suppression test (ie Overnight, or Liddle's test)

Figure 2: need to crop image (remove lateral markings). Also mention if post contrast scan

Table 2. Immunophenotype: Synaptophysin and Chromogranin A (to be written in English)

Line 86: did you give the patient post operative steroid replacement due to suppression of Pituitary ACTH? Why did you do a SST so early at 6 weeks, as this only telling you about adrenal function, rather than ACTH pituitary reserve. The patient needed a post-operatively 24 urinary cortisol.

I would omit the results of SST, as it doesn't add anything.

Paragraph 89: English needs to be revised

Line 186: English revision

Line 178: the patient's outcome needs to included in the case report, no given in the discussion

Conclusion: needs to be rewritten, as the message really is that this is a straight forward case of ectopic ACTH (ie rapidly progressive, hypokalemia, massive mass in chest), but that Thymic cancer is a rare but catastrophic malignancy with a guarded prognosis.